# The protease corin regulates electrolyte homeostasis in eccrine sweat glands

Meiling He[1,2], Tiantian Zhou[1], Yayan Niu[1,3], Wansheng Feng[1], Xiabing Gu[1,3], Wenting Xu[4], Shengnan Zhang[1,3], Zhiting Wang[1], Yue Zhang[1], Can Wang [1], Liang Dong[1], Meng Liu[1], Ningzheng Dong[1,3]*, Qingyu Wu [1,5]*

**1** Cyrus Tang Hematology Center, Collaborative Innovation Center of Hematology, State Key Laboratory of Radiation Medicine and Prevention, the First Affiliated Hospital, Soochow University, Suzhou, China, **2** Department of Nephrology, the People's Hospital of Suzhou New District, Suzhou, China, **3** MOH Key Laboratory of Thrombosis and Hemostasis, Jiangsu Institute of Hematology, the First Affiliated Hospital of Soochow University, Suzhou, China, **4** International Peace Maternity and Child Health Hospital of China Welfare Institute, Shanghai, China, **5** Cardiovascular & Metabolic Sciences, Lerner Research Institute, Cleveland Clinic, Cleveland, United States of America

* ningzhengdong@suda.edu.cn (ND); wuq@ccf.org (QW)

**Data Availability Statement:** All data are presented in the manuscript and supporting information, including S1 Data set.

**Funding:** This work was supported in part by grants from the National Natural Science

## Abstract

Sweating is a basic skin function in body temperature control. In sweat glands, salt excretion and reabsorption are regulated to avoid electrolyte imbalance. To date, the mechanism underlying such regulation is not fully understood. Corin is a transmembrane protease that activates atrial natriuretic peptide (ANP), a cardiac hormone essential for normal blood volume and pressure. Here, we report an unexpected role of corin in sweat glands to promote sweat and salt excretion in regulating electrolyte homeostasis. In human and mouse eccrine sweat glands, corin and ANP are expressed in the luminal epithelial cells. In corin-deficient mice on normal- and high-salt diets, sweat and salt excretion is reduced. This phenotype is associated with enhanced epithelial sodium channel (ENaC) activity that mediates $Na^+$ and water reabsorption. Treatment of amiloride, an ENaC inhibitor, normalizes sweat and salt excretion in corin-deficient mice. Moreover, treatment of aldosterone decreases sweat and salt excretion in wild-type (WT), but not corin-deficient, mice. These results reveal an important regulatory function of corin in eccrine sweat glands to promote sweat and salt excretion.

## Introduction

A constant normal body temperature is essential in health. In humans, eccrine sweat gland–mediated sweating is critical for preventing overheating [1,2]. Among all mammals, humans have the highest density of eccrine sweat glands [3,4], offering an evolutionary advantage to survive in hot environments and to engage in strenuous physical activities such as hunting and long-distance running [5–7].

The initial sweat produced in eccrine glands is isotonic to plasma. Ion reabsorption in the eccrine glands prevents salt loss and electrolyte imbalance. Despite the vital importance of such a basic skin function, how ion excretion and reabsorption are regulated in eccrine sweat glands remains poorly understood.

Foundation of China (81873840, 81570457 to N.D. and 31500636 to T.Z.) and the Priority Academic Program Development of Jiangsu Higher Education Institutes to Soochow University. The funders had no role in study design, data collection and analysis, decision to publish, or preparation of the manuscript.

**Competing interests:** The authors have declared that no competing interests exist.

**Abbreviations:** ANP, atrial natriuretic peptide; CFTR, cystic fibrosis transmembrane conductance regulator; ECSU, Ethics Committee of Soochow University; ENaC, epithelial sodium channel; Gapdh, glyceraldehyde 3-phosphate dehydrogenase; hcKO, heart-conditional corin KO; HRP, horseradish peroxidase; KO, knockout; NPR-A, natriuretic peptide receptor-A; RT-PCR, reverse transcription PCR; SMA, smooth muscle actin; WT, wild-type.

Corin is a protease that activates atrial natriuretic peptide (ANP), a hormone that regulates salt–water balance via its receptor, natriuretic peptide receptor-A (NPR-A) [8,9]. In addition to the heart, corin is also expressed in the kidney, where corin and ANP were reported to inhibit epithelial sodium channel (ENaC) that is essential for sodium reabsorption [10–12]. In mouse and tiger skins, corin has been shown to regulate coat color in an agouti-dependent pathway [13–15]. In humans, corin-positive cells were found in dermal progenitor populations [16]. Deleterious *CORIN* variants were also identified in individuals with hypertension [17–20]. However, there is no evidence to indicate a role of corin in determining human hair color.

In this study, we examined dermal corin expression in humans and mice. Unexpectedly, we detected corin and ANP expression in the luminal epithelial cells of eccrine sweat glands. We studied sweat and ion excretion in wild-type (WT) and corin knockout (KO) mice. Our results indicate that corin plays an important role in the eccrine sweat gland to promote sweat and ion excretion.

## Results

### Corin expression in human skin

Consistent with previous findings in mice [14], we detected corin staining in human hair follicles by immunostaining (S1 Fig). Unexpectedly, we also found corin staining in both secretory and ductal epithelial cells of eccrine sweat glands (Fig 1A and 1B). In the secretory coil portion, corin expression was in the luminal epithelial cells, but not the peripheral myoepithelial cells that were α-smooth muscle actin (SMA) positive (Fig 1C).

Corin converts pro-ANP to ANP, which, in turn, activates its receptor, NPR-A [9,21]. We next examined ANP and NPR-A expression in human eccrine sweat glands. (Here, the term "ANP" refers to both pro-ANP and ANP, as the antibody used did not distinguish pro-ANP from mature ANP). By immunohistochemistry, we found ANP and NPR-A staining in the epithelial cells of human eccrine sweat glands (Fig 1D). These results suggest a potential role of corin and ANP in eccrine sweat glands.

### Corin, ANP, and NPR-A expression in mouse eccrine sweat glands

We next examined corin expression in mouse footpad eccrine sweat glands (Fig 1E). By reverse transcription PCR (RT-PCR) (Fig 1F) and western blotting (Fig 1G), we detected corin mRNA and protein in footpads and hearts (positive control), but not livers (negative control), in WT mice. In corin KO mice (S2 Fig), corin expression was not found in the tissues examined (Fig 1F and 1G). By immunohistochemistry, corin, ANP, and NPR-A expression was detected in eccrine gland epithelial cells in WT mouse footpads (S3A–S3C Fig). In co-immunofluorescent staining, corin expression was found in the luminal epithelial cells but not the SMA-positive peripheral myoepithelial cells (Fig 1H). In the luminal epithelial cells, corin staining overlapped with those of ANP (Fig 1I) and NPR-A (Fig 1J). These results are consistent with the findings in human skins, suggesting that corin and ANP may have a function in eccrine sweat glands.

### Reduced sweat excretion in corin KO mice

To test this hypothesis, we did histological analysis in footpads and found no morphological differences regarding eccrine sweat gland structure and numbers between WT and corin KO mice (S3D Fig). We next did an iodine–starch test [22, 23] with pilocarpine (a muscarinic receptor agonist that stimulates sweat excretion) injection to examine sweat response (S3E Fig). At 1 min after pilocarpine injection, similar numbers of black dots were found in footpads between WT and corin KO mice ($7.7 \pm 0.3$ versus $7.6 \pm 0.3$ /mm$^2$ of paw skin, $n = 10$; $P = 0.969$) (S3F and S3G Fig).

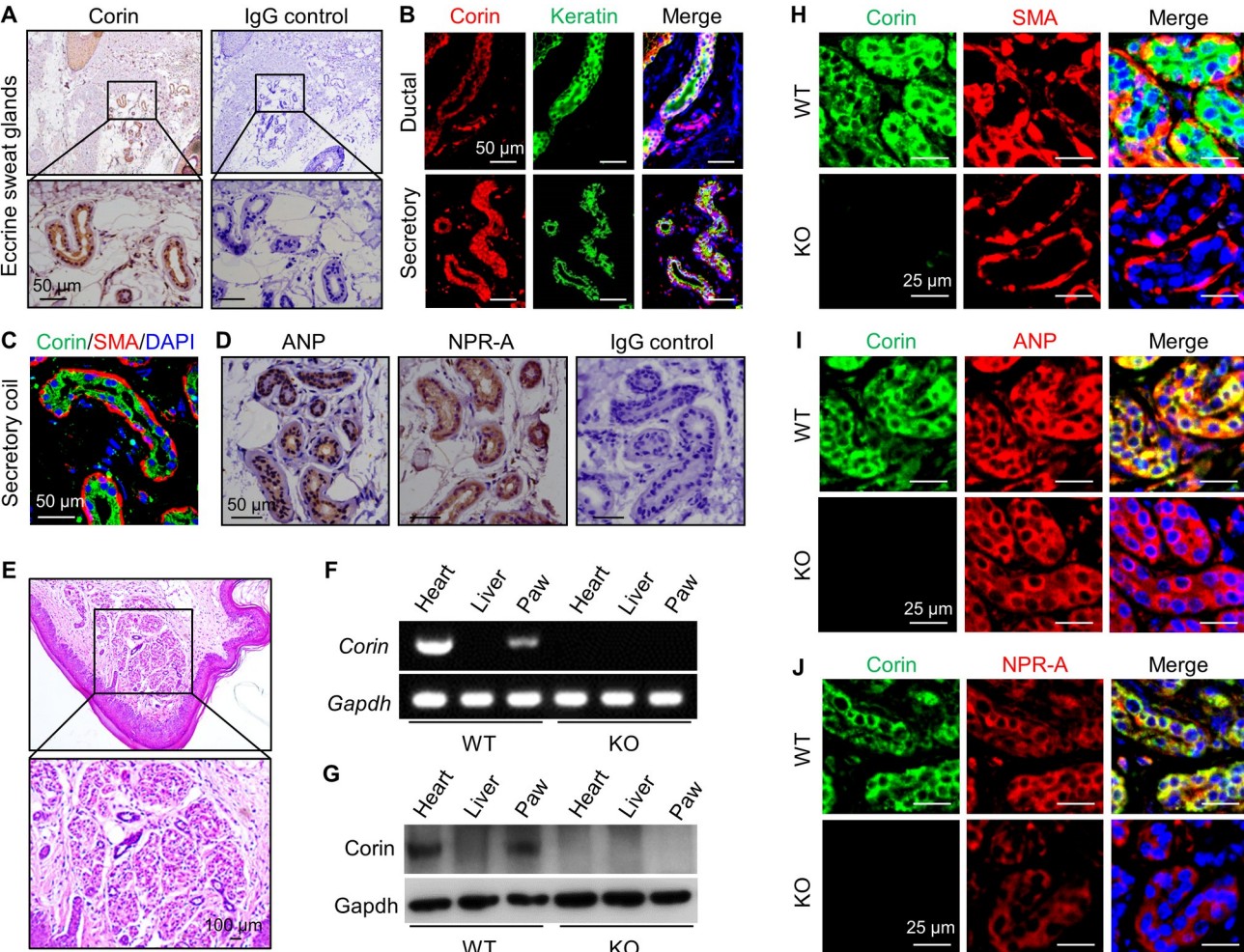

**Fig 1. Corin, ANP, and NPR-A expression in human and mouse eccrine sweat glands.** (A) Immunohistochemical staining of corin (brown) in human eccrine sweat glands. IgG was used in negative controls. (B) Co-staining of corin (red) and keratin (an epithelial marker) (green) in ductal and secretory epithelial cells of human eccrine sweat glands. (C) Co-staining of corin (green) and SMA (red) in the secretory epithelial cells. (D) Immunohistochemical staining of ANP and NPR-A (brown) in human eccrine sweat glands. (E) HE staining of a mouse paw skin section with eccrine sweat glands. (F and G) RT-PCR (F) and western blotting (G) analysis of corin expression in WT and corin KO mouse footpads. Hearts (positive) and livers (negative) were controls. (H–J) Co-staining of corin (green) and SMA (H), ANP (I), or NPR-A (J) (red) in eccrine sweat glands from WT and corin KO mouse footpads. Scale bars are indicated. Data are representative of at least 3 experiments in each set of the studies. ANP, atrial natriuretic peptide; HE, hematoxylin–eosin; IgG, immunoglobulin G; KO, knockout; NPR-A, natriuretic peptide receptor-A; RT-PCR, reverse transcription PCR; SMA, smooth muscle actin; WT, wild-type.

Compared to WT mice, however, corin KO mice had reduced sweat excretion, as indicated by smaller black-staining areas in footpads, analyzed at 2 min after pilocarpine injection ($3.0 \pm 0.3$ versus $4.6 \pm 0.3\%$ of paw skin area, $n = 14$ to $17$; $P < 0.001$) (Fig 2A and 2B). Using a digital camera method to examine the number and size of sweat droplets and calculate sweat volume [24], we found reduced sweat volumes in corin KO mice compared to those in WT mice ($3.1 \pm 0.1$ versus $4.0 \pm 0.1$ nL/mm$^2$ of paw skin, $n = 7$ to $8$; $P < 0.001$) (Fig 2C and 2D), indicating that corin deficiency impairs sweat excretion.

To understand if the role of corin in sweat excretion is mediated by the endocrine function of cardiac corin or a local function of corin in the skin, we examined sweat excretion in heart-conditional corin KO (hcKO) mice, in which the *Corin* gene was disrupted specifically in the heart (Fig 2E). In the iodine–starch test, we found similar sweat excretion in WT and corin

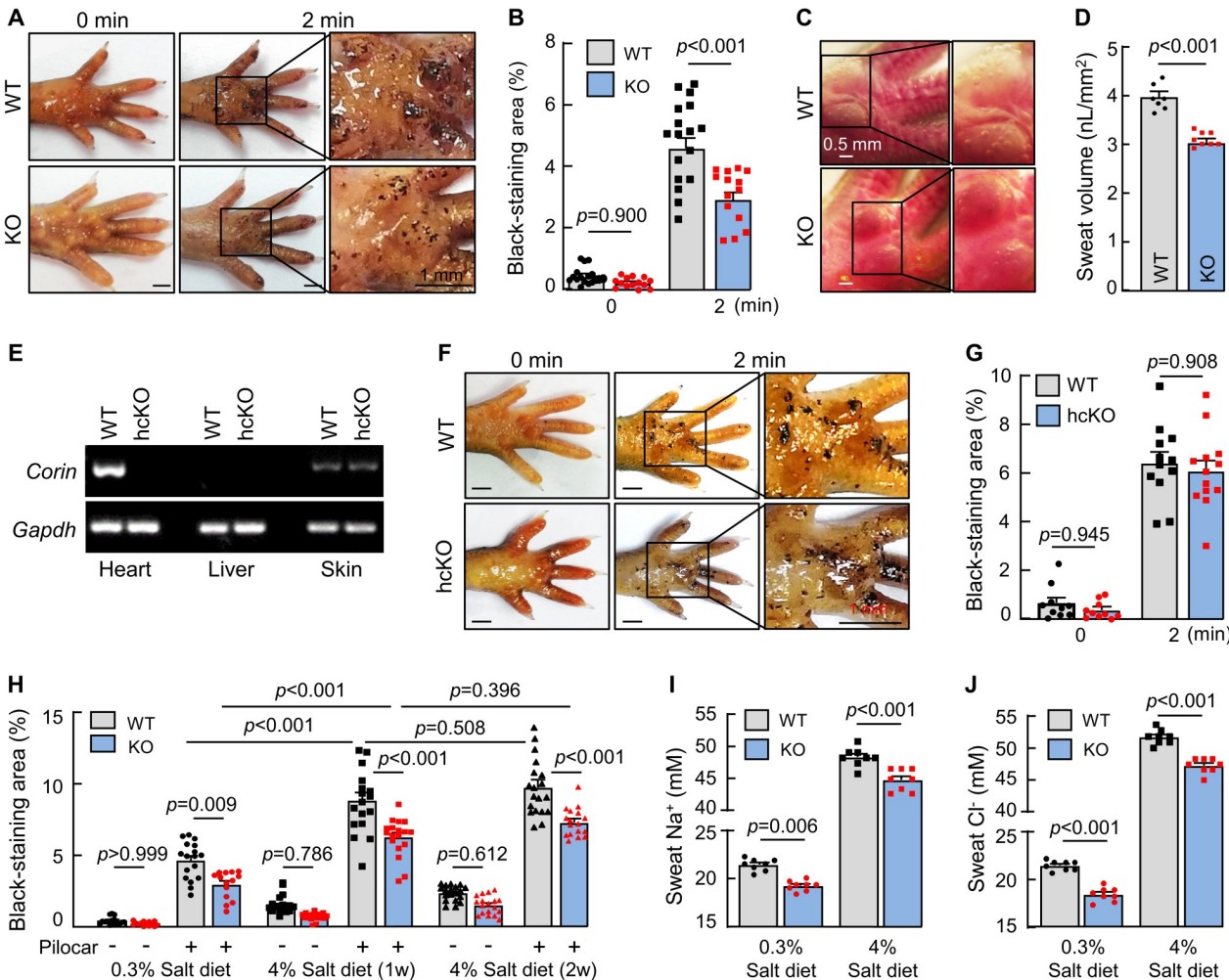

**Fig 2. Sweat response in WT, corin KO, and corin hcKO mice.** (A and B) An iodine–starch assay was used to measure sweat response in WT and corin KO mice. Photos were taken before (0 min) and 2 min after pilocarpine (pilocar) injection. Sweat excretion is indicated by black staining. Areas of black staining were analyzed by computer software. Quantitative data are presented. (C and D) To calculate sweat volume, photos were taken by a digital camera 10 min after pilocarpine injection. The number and diameter of sweat droplets were analyzed by computer software. Representative photos and quantitative data are presented. (E) RT-PCR analysis of corin expression in hearts, livers, and footpads (skin) from WT and hcKO mice. (F and G) Sweat excretion in WT and corin hcKO mice was analyzed by the iodine–starch assay. Representative photos and quantitative data are shown. (H) Sweat excretion in WT and corin KO mice on 0.3% or 4% salt diet before (−) and 2 min after (+) pilocarpine (pilocar) injection. (I and J) Sweat Na$^+$ (I) and Cl$^-$ (J) levels in WT and corin KO mice on 0.3% or 4% salt diet and with pilocarpine injection. In bar graphs, data are mean ± SEM. Each dot represents data from 1 mouse, except for sweat Na$^+$ and Cl$^-$ data, in which each dot represents data from 8 paws of 2 mice. *P* values were analyzed by Student *t* test or 1-way ANOVA, and the original numerical values are in S1 Data. hcKO, heart-conditional corin KO; KO, knockout; RT-PCR, reverse transcription PCR; WT, wild-type.

hcKO mice (6.4 ± 0.5 versus 6.1 ± 0.4% of paw skin area, *n* = 12 to 13; *P* = 0.908) (Fig 2F and 2G), indicating that cardiac corin is unnecessary for sweat excretion.

## Reduced sweat and Na$^+$ and Cl$^-$ excretion in corin KO mice on normal- and high-salt diets

In humans, sweat Na$^+$ levels positively correlate with dietary salt intakes [25–27]. We examined sweat and ion (Na$^+$ and Cl$^-$) excretion in WT and corin KO mice on normal- (0.3% NaCl) and high-salt (4% NaCl) diets. Increased sweat excretion was found in WT and corin KO mice on high-salt diet (Fig 2H). Compared to those in WT mice, levels of sweat excretion

were approximately 25% to 35% lower in corin KO mice on both normal- and high-salt diets. Sweat Na$^+$ (Fig 2I) and Cl$^-$ (Fig 2J) levels were also approximately 7% to 14% lower in corin KO mice, whether on the normal- or high-salt diet, supporting a role of corin in promoting sweat and ion excretion.

## Effects of amiloride on sweat and Na$^+$ excretion

In the kidney, ANP inhibits ENaC activity to increase natriuresis [28,29]. In humans, ENaC mediates Na$^+$ reabsorption in eccrine sweat glands [30]. By immunohistochemistry (Fig 3A) and co-immunofluorescent staining (Fig 3B), we detected eccrine gland epithelial β-ENaC expression in WT and corin KO mice. Levels of *Scnn1b* (encoding β-ENaC) mRNA and β-ENaC protein were similar in footpads between WT and corin KO mice, as assessed by quantitative RT-PCR and western blotting, respectively (S4A and S4B Fig). We treated the mice with amiloride, an ENaC inhibitor. In corin KO mice on normal- and high-salt diets, sweat excretion increased to similar levels in WT mice after amiloride treatment (Fig 3C and 3D). Amiloride treatment also normalized sweat Na$^+$ (Fig 3E) and Cl$^-$ (Fig 3F) levels between WT and corin KO mice. These results indicate that enhanced ENaC activity is likely responsible for the reduced sweat and Na$^+$ excretion in corin KO mice.

Corin KO mice develop salt-sensitive hypertension [12]. As reported previously [12] and verified in this study (S4C Fig), amiloride treatment lowered blood pressure in corin KO mice on the high-salt diet. To exclude the possibility that the observed effect of amiloride on sweat excretion in corin KO mice was caused indirectly by blood pressure lowering, we treated WT and corin KO mice with amlodipine, a calcium channel blocker that lowers blood pressure [12], which decreased blood pressures in WT and corin KO mice to comparable levels (S4D Fig). However, sweat excretion remained low in corin KO mice without or with amlodipine treatment (S4E Fig), indicating that sweat excretion is not directly linked to systemic blood pressure levels in our experiments.

## Effects of CFTR inhibition on sweat and Cl$^-$ excretion

In addition to reduced sweat Na$^+$ levels, reduced sweat Cl$^-$ levels were observed in corin KO mice. In humans, cystic fibrosis transmembrane conductance regulator (CFTR) is the primary Cl$^-$ channel in eccrine sweat glands [31]. Consistently, we detected epithelial CFTR expression in eccrine sweat glands in WT and corin KO mice (Fig 3G and 3H). In quantitative RT-PCR, similar *Cftr* mRNA levels were found in footpads between WT and corin KO mice (S4F Fig). When corin KO mice were treated with CFTR(inh)-172, a CFTR inhibitor [32], sweat excretion increased and became comparable to that in similarly treated WT mice on the normal- and high-salt diets (Fig 3I and 3J). Sweat Cl$^-$ (Fig 3K) and Na$^+$ (Fig 3L) levels also became comparable between WT and corin KO mice, indicating that CFTR activity is increased in corin KO mouse eccrine sweat glands.

## Effects of aldosterone on sweat and Na$^+$ and Cl$^-$ excretion

Aldosterone promotes Na$^+$ reabsorption in eccrine sweat glands by enhancing ENaC activity [33]. To test if corin antagonizes aldosterone in promoting sweat and ion excretion, we treated WT and corin KO mice with aldosterone and examined sweat and Na$^+$ and Cl$^-$ excretion. Decreased sweat excretion was observed in WT, but not corin KO, mice on normal- and high-salt diets, resulting in similar sweat excretion levels between WT and corin KO mice (Fig 4A and 4B). Sweat Na$^+$ (Fig 4C) and Cl$^-$ (Fig 4D) levels also became similar between WT and corin KO mice treated with aldosterone. These results support a role of corin in antagonizing aldosterone, thereby promoting sweat and salt excretion in eccrine glands.

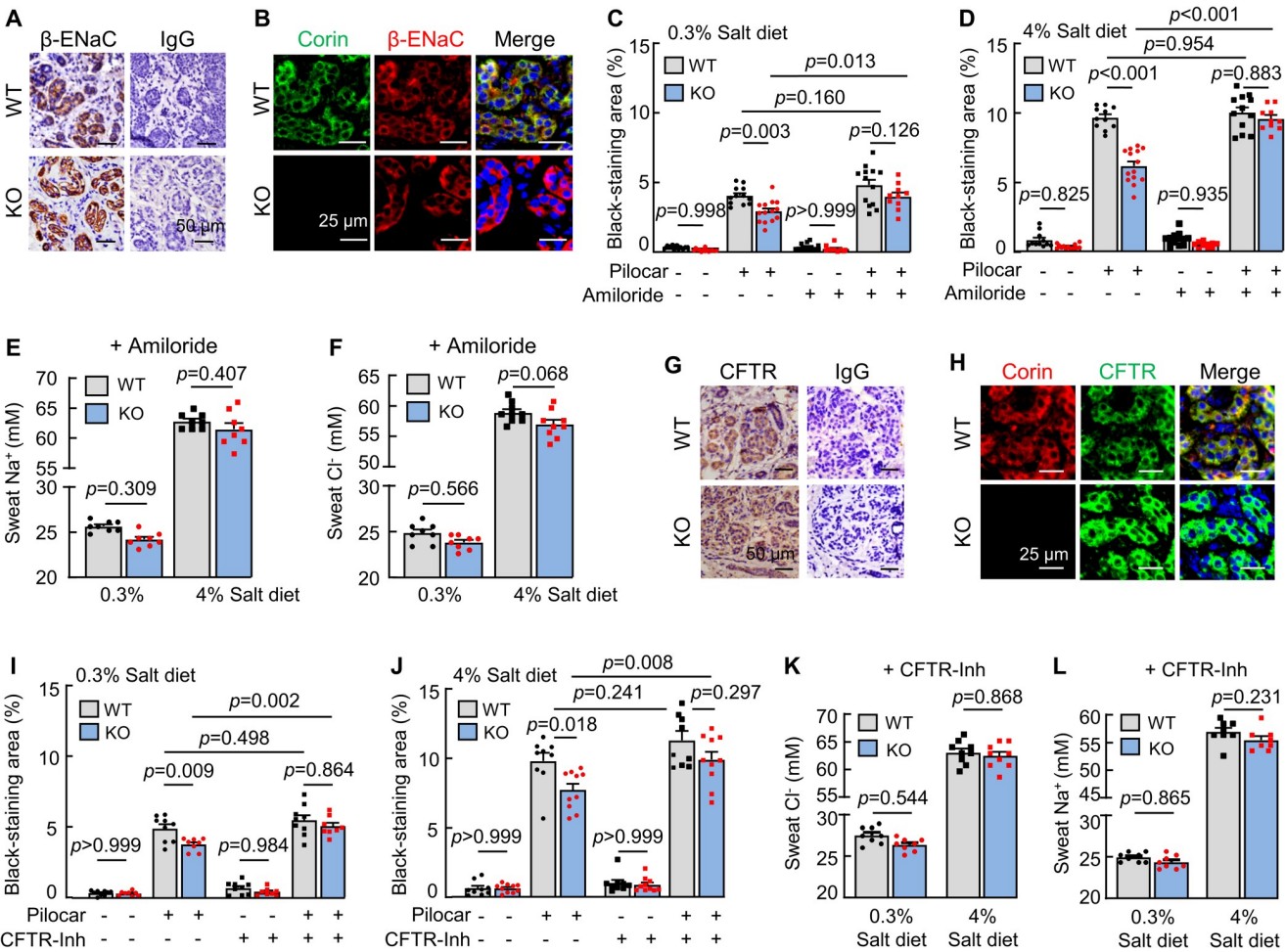

**Fig 3. Effects of ENaC and CFTR inhibitors on sweat excretion.** (A) Immune staining of β-ENaC (brown) in eccrine sweat glands from WT and corin KO mouse footpads. IgG was a negative control. (B) Co-staining of corin (green) and β-ENaC (red) in WT and corin KO eccrine sweat glands. (C and D) Sweat excretion, indicated by black-staining areas, was measured in WT and corin KO mice on 0.3% (C) or 4% (D) salt diet, with (+) or without (−) amiloride treatment, before (−) and 2 min after (+) pilocarpine (pilocar) injection. (E and F) Sweat $Na^+$ (E) and $Cl^-$ (F) levels in WT and corin KO mice on 0.3% or 4% salt diet and treated with amiloride. (G) Immune staining of CFTR (brown) in eccrine sweat glands in WT and corin KO mouse footpads. (H) Co-staining of corin (red) and CFTR (green) in eccrine sweat glands in WT and corin KO mouse footpads. (I and J) Sweat excretion, indicated by black-staining areas, was measured in WT and corin KO mice on 0.3% (I) or 4% (J) salt diet, with (+) or without (−) CFTR(inh)-172 (CFTR-Inh) treatment, before (−) and 2 min after (+) pilocarpine (pilocar) injection. (K and L) Sweat $Cl^-$ (K) and $Na^+$ (L) levels in WT and corin KO mice on 0.3% or 4% salt diet and treated with (+) CFTR-Inh. In bar graphs, data are mean ± SEM. Each dot represents data from 1 mouse, except for sweat $Na^+$ and $Cl^-$ data, in which each dot represents data from 8 paws of 2 mice. P values were analyzed by 1-way ANOVA, and the original numerical values are in S1 Data. CFTR, cystic fibrosis transmembrane conductance regulator; ENaC, epithelial sodium channel; IgG, immunoglobulin G; KO, knockout; WT, wild-type.

## Discussion

In this study, we examined corin expression in the skin. In agreement with previous reports in mice and tigers [13–15,34], we detected corin expression in human hair follicles. It remains unclear if corin plays a role in hair growth and/or pigmentation in humans. Unexpectedly, we detected corin expression in the luminal epithelial cells of human and mouse eccrine sweat glands. As a transmembrane protease [35], corin is expected to function at expression sites. In supporting this hypothesis, we found reduced sweat and salt excretion in corin KO mice on normal- and high-salt diets. This defect was associated with enhanced ENaC activity, as indicated by the studies with amiloride, which normalized sweat and salt excretion in corin KO

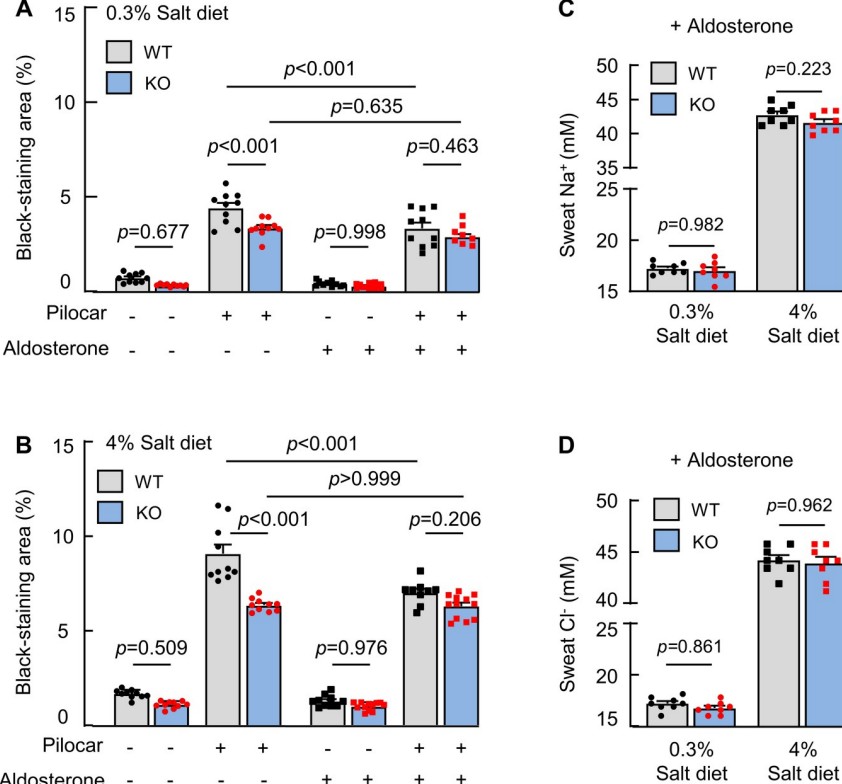

**Fig 4. Effects of aldosterone on sweat and salt excretion.** (A and B) Quantitative data (mean ± SEM) of black-staining areas in the iodine–starch test in WT and corin KO mice on 0.3% (A) or 4% (B) salt diet without (−) or with (+) aldosterone treatment and pilocarpine (pilocar) injection. (C and D) Sweat $Na^+$ (C) and $Cl^-$ (D) levels in WT and corin KO mice on 0.3% or 4% salt diet and treated with (+) aldosterone. Data are mean ± SEM. Each dot represents data from 1 mouse, except for sweat $Na^+$ and $Cl^-$ data, in which each dot represents data from 8 paws of 2 mice. *P* values were analyzed by 1-way ANOVA, and the original numerical values are in S1 Data. KO, knockout; WT, wild-type.

mice on normal- and high-salt diets. ANP is known to inhibit ENaC activity via NPR-A signaling [21,28,29,36]. In eccrine sweat glands, ENaC mediates $Na^+$ reabsorption [30]. Consistently, we detected overlapping corin, ANP, and NPR-A expression in the epithelial cells of eccrine sweat glands. Possibly, the lack of corin in the KO mice prevents ANP activation and hence ENaC inhibition, resulting in increased ENaC activity and reduced sweat and salt excretion. Previously, we and others reported a potential role of corin and ANP in the kidney to inhibit ENaC activity, thereby promoting salt and urine excretion [10,12]. The results in this study indicate that a similar corin–ANP-mediated ENaC inhibitory mechanism operates independently in the eccrine sweat glands to regulate sweat and salt excretion. Additional studies with eccrine sweat gland–specific KO mice will be important to verify our results.

The natriuretic peptides were evolved in early vertebrates to regulate salt–water balance [37,38]. In marine birds, nasal salt glands expel excessive salt [39]. Similar nasal salt glands are preserved in aquatic birds. ANP antigen was detected in the nasal salt glands in Pekin ducks [40], in which circulating ANP was shown to modulate the angiotensin–aldosterone system [41]. In humans, ANP is part of the cardiac endocrine function [42,43]. Previously, ANP-like immunoreactivity was reported in the nerves around human eccrine sweat glands, but not the epithelial cells in the glands [44]. NPR-A (also called guanylyl cyclase-A) activity was also detected in epithelial cell membranes of human eccrine sweat glands [45].

In this study, we found reduced sweat and salt excretion in corin KO mice, but not in corin hcKO mice that lacked cardiac corin, indicating that the circulating ANP generated by cardiac corin in unnecessary for sweat and salt excretion in the skin. These results suggest that the corin function in the skin, likely mediated by local ANP activation, is part of an autocrine mechanism in promoting salt excretion in mammalian eccrine sweat glands. Salt reabsorption occurs mostly in the sweat duct. At this time, the significance of corin expression in the sweat secretory portion remains to be determined. Previously, intradermal introduction of exogenous ANP, through microdialysis fibers, into the interstitial fluid did not alter sweating and cutaneous vasodilatation in men [46]. The reason for the lack of response is unclear. Possibly, exogenous ANP in the interstitial space cannot reach the eccrine sweat gland lumen and the microvasculature to activate NPR-A on the surface of epithelial and endothelial cells [46]. Further studies are needed to verify such a possibility.

CFTR mediates sweat $Cl^-$ reabsorption. In patients with cystic fibrosis, *CFTR* mutations cause high sweat $Cl^-$ levels, which, in turn, inhibit ENaC-mediated $Na^+$ reabsorption [31]. In corin KO mice, both sweat $Na^+$ and $Cl^-$ levels were reduced. Amiloride or CFTR(inh)-172 treatment increased sweat $Na^+$ and $Cl^-$ levels in corin KO mice, reflecting the functional coupling of $Na^+$ and $Cl^-$ regulation in eccrine sweat glands [30]. Previously, exogenous ANP treatment up-regulated intestinal CFTR expression in rats and stimulated CFTR activity in *Xenopus* oocytes [47,48]. In our study, we found similar *Scnn1b* (encoding β-ENaC) and *Cftr* mRNA levels in footpads between WT and corin KO mice, but increased sweat gland ENaC and CFTR activities in corin KO mice. These results indicate that the function of endogenous corin and ANP in mouse footpads is mediated primarily by regulating ENaC and CFTR activity but not expression levels. Further studies are required to understand if the apparent different results in rat intestines and *Xenopus* oocytes were due to different experimental settings.

In sweat glands, aldosterone-enhanced ENaC activity and $Na^+$ reabsorption are important for salt conservation [33]. Physiologically, hormonal actions are tightly controlled. To date, how aldosterone action in sweat glands is counterbalanced remains unknown. We showed that aldosterone treatment reduced sweat excretion in WT mice but not corin KO mice. Possibly, salt excretion promoted by corin and salt reabsorption promoted by aldosterone were in equilibrium in WT mice. Exogenous aldosterone treatment tilted the balance, increasing salt reabsorption and reducing sweat and salt excretion. In corin KO mice, on the other hand, endogenous aldosterone apparently reached the maximal effect in sweat glands. As a result, exogenous aldosterone treatment had little effect on sweat and salt excretion in corin KO mice. These results indicate that corin functions as an aldosterone-antagonizing mechanism in eccrine sweat glands. Additional studies will be important to verify this hypothesis.

In summary, sweating is an important skin function. Regulation of $Na^+$ and $Cl^-$ excretion in sweat glands prevents excessive salt loss and electrolyte imbalance. Corin is known for its role in the cardiac endocrine function to regulate blood volume and pressure. By studying human skin tissues and mouse models, we have uncovered a crucial role of corin in eccrine sweat glands to promote salt and sweat excretion. Our results indicate that this corin function serves as an aldosterone-antagonizing mechanism in eccrine sweat glands. These findings provide new insights into the physiological mechanism that regulates skin function.

## Materials and methods

### Human tissues

Normal scalp tissues from anonymous donors were from a biobank from the Pathology Department, which was approved by the Ethics Committee of Soochow University (ECSU-

201800052) and conducted according to the principles of expressed in the Declaration of Helsinki. The experiments were done according to the approved guidelines.

## Mice

Experiments in WT C57BL/6 mice and corin KO and hcKO mice, as described in S1 Text, were approved by the Animal Use and Care Committee of Soochow University (201708A105) and conducted in accordance with the guidelines for the ethical treatment and handling of animals in research.

## Immune staining

Paraformaldehyde-fixed tissues in paraffin were cut and stained with primary antibodies against corin, pro-ANP/ANP, NPR-A, β-ENaC, CFTR, SMA, and cytokeratin 18. Horseradish peroxidase (HRP)-conjugated or Alexa 488 (green)- or 594 (red)-labeled secondary antibodies were used for immunohistochemistry and immunofluorescent staining, respectively. Experimental details are described in S1 Text. Stained sections were examined with light (Leica DM2000 LED, Leica Microsystems, Wetzlar, Germany) and confocal (Olympus FV1000, Olympus, Tokyo, Japan) microscopes.

## RT-PCR

RT-PCR and quantitative PCR were used to analyze *Corin*, *Scnn1b* (encoding β-ENaC), *Cftr*, *Npr1*, and *Pcsk6* expression in mouse tissues. *Glyceraldehyde 3-phosphate dehydrogenase* (*Gapdh*) expression was analyzed in parallel as a control. Experimental conditions and primer sequences are described in the Supporting information Methods and Table A in S1 Text, respectively.

## Western blotting

Corin protein in tissues was examined by western blotting. Tissues were homogenized in 1% Triton X-100 (v/v), 50 mM Tris-HCl, pH 8.0, 150 mM NaCl, and a protease inhibitor mixture (Roche Diagnostics, Indianapolis, Indiana, United States of America, 1:100 dilution). Proteins were quantified using a BCA protein assay (Thermo Fisher Scientific, Waltham, Massachusetts, USA) and analyzed by SDS-PAGE and western blotting with an anti-corin antibody (1:1,000 dilution) [17] or anti-β-ENaC antibody (1:1,000 dilution).

## Sweat response

An iodine–starch method [22,23] was used to examine sweat response. Mice were anesthetized. A hind paw was painted with an iodine solution (Fluka, Ronkonkoma, New York, USA) and starch (Adamas, Emeryville, California, USA) in mineral oil, as described in S1 Text. Pilocarpine (5 mg/kg of body weight), an agonist that acts on muscarinic receptor M3 subtype to stimulate sweat excretion, was injected, s.c. When sweat is excreted and encounters iodine–starch, it turns into black color. At different times, photos were taken. Black dot counts (for eccrine sweat gland numbers at 1 min) and black-staining areas (for sweat excretion at 2 min) were quantified using Image-Pro-Plus software. To measure sweat volume, a stereomicroscope (Olympus, SZX16) and digital camera method were used [24]. A hind paw was immersed in water-saturated mineral oil. At 10 min post-pilocarpine injection, photos were taken, and sweat droplet numbers and diameters were quantified by Image-Pro-Plus software to calculate sweat volume.

### Sweat Na$^+$ and Cl$^-$

Mice were injected with pilocarpine. After 10 min, sweat droplets on footpads immersed in water-saturated mineral oil were mixed with double distilled water (30 μL/per 8 paws) and collected. Na$^+$ and Cl$^-$ concentrations were measured by an electrolyte analyzer (Shelf Scientific, Los Angeles, California, USA, 6230M). Data from each 30-μL diluted sweat collection were recorded as 1 data point.

### Treatment of ENaC and CFTR inhibitors and aldosterone

Mice (males and females, 8 to 10 weeks old) were fed 0.3 or 4% NaCl diet. After 2 weeks, the mice were treated with amiloride (ENaC inhibitor) (Abcam, Cambridge, Massachusetts, USA, 3 mg/kg of body weight, i.p.; daily × 5), CFTR(inh)-172 (CFTR inhibitor) (MedChemExpress, Monmouth Junction, New Jersey, USA, 0.3 mg/kg of body weight, i.p.; daily × 3), or aldosterone (APExBIO, Houston, Texas, USA, 0.2 mg/kg of body weight, i.p.; daily × 3). After the treatment, sweat responses and levels of Na$^+$ and Cl$^-$ were analyzed.

### Blood pressure

A computerized tail-cuff system (Visitech Systems, Apex, North Carolina, USA, BP-2000) was used to measure blood pressure [49]. Mice were acclimated to the instrument. Tails were inserted in the cuff for blood pressure measurements, which included 5 preconditioning cycles and 20 regular cycles with 5 s between 2 cycles and maximal cuff pressure of 150 mmHg.

### Statistical analysis

Data were analyzed using SPSS 17.0 and Prism 8.0 (GraphPad) software. Two-tailed Student $t$ test was used to compare 2 groups if the data passed the normality and equal variance tests. If the data did not pass the normality or equal variance test, Mann–Whitney test was used for 2 independent sample comparisons. One-way ANOVA followed by Tukey post hoc analysis was used to compare 3 or more groups. Data are presented as means ± SEM. $P$ values of $<0.05$ were considered to be significant.

### Supporting information

**S1 Fig. Corin expression in human hair follicles.** (A) Immunohistochemical analysis of corin expression in human scalp sections. Positive corin staining (brown) was detected in hair follicles. A boxed area is shown below in a higher magnification. Scale bars are indicated. (B) HE, corin, and cytokeratin staining in serial human scalp sections. A normal IgG was used as a negative control in immunohistochemical analysis. Boxed areas are shown below in a higher magnification. Scale bars are indicated. Data are representative of at least 3 experiments. HE, hematoxylin–eosin; IgG, immunoglobulin G.
(PDF)

**S2 Fig. Generation of corin KO mice.** (A) Illustration of the strategy to disrupt the *Corin* gene by inserting 2 loxP sites flanking exon 4. PCR primers used in genotyping are indicated. (B–D) PCR analysis using indicated oligonucleotide primers to identify mice with *Cor*$^{flox}$ and *Cor*$^{del4}$ alleles before (B and C) and after (D) exon 4 was deleted by crossing with mice expressing *Cre*. KO, knockout.
(PDF)

**S3 Fig. Corin, ANP, and NPR-A expression and analysis of eccrine sweat glands in mouse footpads.** (A and B) Immunohistochemical staining of corin (A), ANP, and NPR-A (B) in

footpad sections from WT and corin KO mice. In (A), eccrine sweat glands are indicated by red arrowheads. In (B), positive ANP and NPR-A staining in epithelial cells are indicated by black arrowheads. Normal IgG was used as a negative control. Scale bars are indicated. (C) Levels of *Npr1* mRNA levels in footpads from WT and corin KO mice, analyzed by quantitative RT-PCR. *n* = 8 mice per group. (D) Footpad sections from WT and corin KO mice were stained with HE. Eccrine sweat glands are indicated by red arrowheads. No changes in eccrine sweat gland structure and numbers were observed between WT and corin KO mice. Data are representative of at least 3 experiments. (E) Illustration of the iodine–starch test used in this study. Mouse paws were cleaned and coated with iodine and starch. Pilocarpine, a sweat stimulant, was injected, s.c., in paws. Photos were taken at 0 (control), 1 (for sweat gland numbers), and 2 min (for sweat excretion) after the injection. (F) Representative photos taken at 1 min after pilocarpine injection in WT and corin KO mice. (G) Quantitative data of black-staining areas in WT and corin KO mouse paws are presented as mean ± SEM. *n* = 10 mice per group. Data in C and G were analyzed with 2-tailed Student *t* test and Mann–Whitney test, respectively, and the original numerical values are in S1 Data. ANP, atrial natriuretic peptide; HE, hematoxylin–eosin; IgG, immunoglobulin G; KO, knockout; NPR-A, natriuretic peptide receptor-A; RT-PCR, reverse transcription PCR; WT, wild-type.
(PDF)

**S4 Fig. ENaC and CFTR expression and effects of amiloride and amlodipine on blood pressure and sweat excretion.** (A) Quantitative RT-PCR analysis of *Scnn1b* mRNA, encoding β-ENaC, in footpads from WT and corin KO mice on 0.3% salt diet. (B) Western blotting of β-ENaC protein in footpads from WT and corin KO mice (*n* = 4 per group) on 0.3 salt diet. Proteins expression levels were quantified by densitometric analysis of the western blot. (C) Systolic BP in WT and corin KO mice on 0.3% or 4% salt diet treated with vehicle or amiloride. (D) Systolic BP in WT and corin KO mice on 0.3% salt diet treated with vehicle or amlodipine. (E) Sweat excretion was analyzed by the iodine–starch test in WT and corin KO mice on 0.3% salt diet and treated with (+) or without (−) amlodipine and pilocarpine (pilocar). (F) Quantitative RT-PCR analysis of *Cftr* mRNA, encoding CFTR, in footpads from WT and corin KO mice on 0.3% salt diet. All data are mean ± SEM. *P* values were analyzed by 2-tailed Student *t* test (in A, C, and E) or 1-way ANOVA (in B and D). In bar graphs, each dot represents data from 1 mouse, and the original numerical values are in S1 Data. BP, blood pressure; CFTR, cystic fibrosis transmembrane conductance regulator; ENaC, epithelial sodium channel; KO, knockout; RT-PCR, reverse transcription PCR; WT, wild-type.
(PDF)

**S1 Text. Supporting information Materials and methods and Tables A and B.** Detailed experimental procedures are described in the Supporting information Materials and methods. Primers used in the study are listed in Table A. Antibodies used in the study are listed in Table B.
(DOCX)

**S1 Data. Original data used for bar graphs.** Original numerical data underlying bar graphs in Figs 2–4 and S3 and S4 Figs.
(XLSX)

**S1 Original Blots and Gels. Original western blot and gel images.** Raw images used for panels in Figs 1 and 2 and S2 and S4 Figs.
(PDF)

## Author Contributions

**Conceptualization:** Meiling He, Ningzheng Dong, Qingyu Wu.

**Data curation:** Meiling He, Tiantian Zhou, Liang Dong, Ningzheng Dong.

**Formal analysis:** Meiling He, Ningzheng Dong, Qingyu Wu.

**Funding acquisition:** Tiantian Zhou, Ningzheng Dong.

**Investigation:** Meiling He, Tiantian Zhou, Yayan Niu, Wansheng Feng, Xiabing Gu, Wenting Xu, Shengnan Zhang, Zhiting Wang, Yue Zhang, Can Wang, Liang Dong, Meng Liu.

**Methodology:** Meiling He, Yayan Niu, Wansheng Feng.

**Project administration:** Meng Liu, Ningzheng Dong, Qingyu Wu.

**Resources:** Tiantian Zhou, Ningzheng Dong.

**Supervision:** Meng Liu, Ningzheng Dong, Qingyu Wu.

**Validation:** Tiantian Zhou.

**Visualization:** Meiling He, Qingyu Wu.

**Writing – original draft:** Meiling He, Ningzheng Dong, Qingyu Wu.

**Writing – review & editing:** Meiling He, Tiantian Zhou, Yayan Niu, Wansheng Feng, Xiabing Gu, Wenting Xu, Shengnan Zhang, Zhiting Wang, Yue Zhang, Can Wang, Liang Dong, Meng Liu, Ningzheng Dong, Qingyu Wu.

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
