## [Editor Report · Decision Letter 0]

10 Aug 2020

Dear Dr Wu, 

Thank you for submitting your manuscript entitled "The protease corin regulates electrolyte homeostasis in eccrine sweat glands" for consideration as a Research Article by PLOS Biology.

Your manuscript has now been evaluated by the PLOS Biology editorial staff [as well as by an academic editor with relevant expertise] and I am writing to let you know that we would like to send your submission out for external peer review.

Please re-submit your manuscript within four working days, i.e. by August 14th 2020.

IMPORTANT: we note that you submitted this manuscript as a regular Research Article, but we think that it would be better considered as a Short Report. The maximum allowable number of Figures for a Short Report is 4, but we see that you currently have 8. You will need to combine these or move some of the material to the supplementary Figures. Please upload the re-organised Figures and manuscript when you upload the metadata requested in the next paragraph.

Kind regards,

Maya Capelson,

PLOS Biology

---

## [Decision Letter · Decision Letter 1]

15 Oct 2020

Dear Dr Wu,

Thank you very much for submitting your manuscript "The protease corin regulates electrolyte homeostasis in eccrine sweat glands" for consideration as a Short Report at PLOS Biology. Thank you also for your patience as we completed our editorial process, and please accept my apologies for the delay in providing you with our decision. Your manuscript has been evaluated by the PLOS Biology editors, an Academic Editor with relevant expertise, and by four independent reviewers.

The reviews are attached below. You will see that the reviewers find your results interesting and novel and think it is worth pursuing publication of the manuscript in PLOS Biology. Thus we are pleased to offer you the opportunity to address the points raised by the reviewers in a revised version that we anticipate should not take you very long. Regarding the points raised by Reviewer 4, please address in the Discussion the function of this cascade in the kidney - given that renal function is critical in regulating electrolyte balance and cardiovascular function - and how this may interact with the sweat glands to regulate core body temperature, and all the minor issues. Once you resubmit, we will assess your revised manuscript and your response to the reviewers' comments and we may consult the reviewers again.

We expect to receive your revised manuscript within 1 month.

**IMPORTANT - SUBMITTING YOUR REVISION**

*Resubmission Checklist*

*Published Peer Review*

*PLOS Data Policy*

*Blot and Gel Data Policy*

Sincerely,

Ines

--

Ines Alvarez-Garcia, PhD,

Senior Editor,

ialvarez-garcia@plos.org,

PLOS Biology

Reviewers’ comments

Rev. 1:

He and colleagues describe a novel role for the transmembrane protease corin in the regulation of sweat and salt excretion in eccrine sweat glands that parallels previously described roles of corin in renal sodium reabsorption and blood pressure regulation. They observe that corin, its main physiological substrate atrial natriuretic peptide (ANP), and the ANP receptor NPR-A are all expressed in ductal and secretory epithelial cells in human and mouse eccrine sweat glands. In mice, corin deficiency reduces sweat and salt excretion on both normal and high salt diets. Administration of amiloride and aldosterone both eliminate the difference in sweating and salt secretion between corin KO and control mice, suggesting that corin regulates sodium and water reabsorption through ENaC and counteracts the stimulatory effects of aldosterone on this sodium channel. A number of genetic and pharmacological approaches have been used to support this model and provide substantial mechanistic insight for a short report. Extensive and appropriate controls have been performed, and conclusions drawn are fully supported by the data provided. The description of a previously unappreciated and physiologically important regulatory pathway in eccrine sweat gland function in this study represents an important scientific advance and will no doubt spur future research in this area.

Essential revisions requested:

No morphological differences were found between footpads of corin KO mice and controls. The authors should also comment on whether or not histological differences were observed in eccrine sweat gland structure and numbers.

More information should be provided on the heart-conditional corin KO. How was excision efficiency addressed? Did this mouse display hypertension as observed in the global KO? What is known about plasma electrolytes in global and cardiac specific corin KO mice?

Were the experiments littermate controlled? If not, how were wild-type controls generated, where were they bred, and what was their relation to the knockout mice?

Statistical analysis: Were normality tests performed before t-tests? One-way ANOVA is used for some experiments where the effect of both treatment and genotype are assessed. Does each symbol in bar graphs represent a different mouse or were several paws from the same mouse studied or the same mouse studied more than once?

Non-essential revisions suggested:

As pointed out by the authors, sweating is important to prevent overheating. Have they addressed if corin KO mice are impaired in their ability to maintain core body temperature with increasing ambient temperature? Although not essential, such data would corroborate the model.

Rev. 2:

This is a well-done study revealing a new function of corin/ANP in regulating electrolyte homeostasis in sweat glands. The result section flows well and the experiments seem to be executed carefully and the results nicely presented in the figures. The findings establish a new paradigm for corin/ANP in that the entire system including ANP receptor and ENaC is localized to the sweat glands.

However, the interpretation of some of the results need to be carefully reevaluated and more thoroughly discussed. In particular, since the sweat gland phenotype of corin ko mice is pilocarpine-dependent, the authors need to discuss the physiological relevance of their data more rigorously. In addition, the introduction needs improvement.

1. The introduction does not provide the essential information for understanding the work presented. Please expand on corin, ANP and its receptors and what is known about ANP regulating ENaC and the function of ENaC (e.g. in kidney).

2. Page 4: explain why keratin was used (epithelial marker?).

Page 4: "By (not "in") immunohistochemistry".

Page 4: Fig.1J: The NPR-A signal seems much less in the corin KO compared to wt. May be I missed it, but were NPR-A expression levels determined for wt vs ko? This seems important, since NPR-A activity is linked to ENaC activity and the authors need to assure that the corin ko mice have equal levels of NPR-A as the wt.

3. Page 5: please explain the physiologic meaning of the "black dots" and how pilocarpine functions. Very few readers will be familiar with sweat gland physiology and this particular experimenatl system, which is heavily used in the paper. Also, it appears that the observed corin KO phenotype (reduced sweat excretion, Na and Cl) is entirely dependent on the treatment with pilocarpine and is not observed without it. Therefore, a clear explanation on pilocarpine function and mechanism should be provided. The fact that in the absence of pilocarpine no phenotype was observed raises questions about the physiological relevance of the pilocarpine-dependent reductions; therefore, the authors need to explain and comment on this.

4. page Fig.2: The legend should indicate that pilocarpine was used for Fig. 2I and J.

Fig. 2H shows a significant (p 0.009) reduction with normal diet, but the authors only mention reduction with the high salt diet. Why?

Also, the authors should indicate the % reductions here and in other figures as well.

5. Page 6: The authors show that ENaC expression is similar in wt and ko mice, but that ENaC activity is increased in ko. I thought that corin/ANP regulate the transcription of ENaC and that corin-deficiency increases ENaC transcripts (Zhou, Wu Current Hypertension Rep 2012; Theilig, Wu Am.J.Physiol.Renal Physiol. 2015). If true, then the results need some explanation. If the authors conclude that there is another mechanism by which corin/ANP regulates ENaC, then this needs to be explained and discussed.

6. Page 6, Fig. S4B. the effect of amiloride seems minimal compared to amlodipine. Please comment.

7. Page 7, fig. 3C,D: what are the Na and Cl levels w/o aldosterone?

8. Page 7: To open the discussion with a topic that was not experimentally pursued seems a bit odd.

9. Page 9: I don't quite understand the conclusion made in last sentence in 2nd paragraph. To me the conclusion would be that the presented results contradict the studies with exogenous ANP in that reduced ANP in corin ko mice has the same effect as addition of exogenous ANP (i.e. increased CFTR activity). In addition, ANP actually seems to have a role in regulating CTFR activity, since the corin ko have increased CTFR activity. Also, if the CTFR levels are the same in wt and ko, how then does ANP regulate CTFR activity? Please clarify this and discuss it better.

10. Fig.S3. Some labeling of pictures or explanation in figure legend may be helpful for readers unfamiliar with sweat gland/skin histology, e.g. indicate individual sweat glands, point out that staining is in epithelial layer etc.

Rev. 3: Zaid Abassi – this reviewer has waived anonymity

The present study by He et al entitled: "The protease corin regulates electrolyte homeostasis in eccrine sweat glands" addresses the involvement of corin in the regulation of salt excretion by eccrine sweat gland. Corin is a key enzyme in the conversion of pro ANP into active ANP. This enzyme is localized mainly to the membrane of cardiomyocytes, where it plays a major role in the generation of mature ANP. The latter is involved in the sodium homeostasis and regulation and regulation of blood pressure. In addition to the heart, corin is expressed in mouse and tiger skins, where it regulates coat color in an agouti-dependent pathway. However, so far there is no reports concerning the expression of corin in sweat garland or potential role of this enzyme in their regulation. To the best of my knowledge the present study is the first one to detect corin and ANP expression in the luminal epithelial cells of eccrine sweat glands in both mice and humans. Furthermore, the authors shed light on the function of this axis on ion excretion. Specifically, the authors clearly showed that corin in sweat glands promotes sweat and salt excretion as part of regulating electrolyte homeostasis. By applying corin KO mice, the investigators demonstrated that corin-deficient mice exhibited reduced sweat and salt excretion when placed on normal- and high-salt diets. This phenotype is associated with enhanced epithelial sodium channel (ENaC) activity that mediates Na+ and water reabsorption. Treatment of amiloride, an ENaC inhibitor, normalizes sweat and salt excretion in corin-deficient mice. Moreover, treatment of aldosterone decreases sweat and salt excretion in wild-type, but not corin deficient, mice. Collectively, these results reveal an important regulatory function of corin in eccrine sweat glands where it promotes sweat and salt excretion.

This is a well designed and preformed study. The obtained that is well presented and overall it is well written manuscript. The applied methodology includes physiological and molecular analysis and genetically modified mice aimed at addressing the role of corin in regulating sweat gland function. The histological and immunoblots and immunofluorescent images are of high quality and support the study conclusions. However, the study suffers from few limitations which need to be addressed in order to reach to keen conclusions:

1- Corin is a membranal enzyme and is produced by conversion of procorin into corin by PCSK6. It is very essential to examine the status of PCSK6 in the studied models. The presence of PCSK6 may strength the authors conclusion that a whole machinery of ANP production I present in the eccrine sweat glands of experimental and human origin. In addition, it is of interest how high or low salt affects not only corin expression, but also PCSK6.

2- The authors examined the impact of aldosterone on sweat and salt excretion in wild-type and corin deficient mice. I wonder whether spironolactone block these effects. This protocol supposed to validate the observed inhibitory impact of aldosterone on salt and sweat excretion by eccrine sweat glands. Moreover, measurement of aldosterone levels in wild and corin KO mice on low and high salt diet should be measured.

3- It is unclear whether the "n" represents number of analyzed, fields or animals. The number of animals in each studied group should be indicated in both the text and figure legends.

4- I wonder, if the authors have measured Na+ and Cl- levels in the circulation of the studied groups of animals.

Rev. 4:

The authors analyzed the expression pattern and function of Corin in sweat glands. Corin has not been reported in sweat glands, and thus this report is of interest for people in the exocrine or skin biology fields. The manuscript is well written. However, the major concern is that the function of Corin/ANP/ENaC cascade was established in the kidneys (Polzin D, 2010; Klein J, 2010), and the cascade displayed a similar function in sweat glands. Therefore, the novelty of this report is limited.

I have additional comments as follows.

1. Introduction is not informative. Rather than emphasizing Corin function in hair color, the authors should introduce Corin/ANP/ENaC function in the kidneys. Also, the similarities and dissimilarities of Corin function in kidney and sweat glands should be compared in the Discussion section.

2. In the present study, the authors analyzed the whole-body Corin KO mice, but not skin-specific KO mice. The whole-body KO mice have previously shown retention of sodium and water because of altered kidney function, which may affect sweat secretion directly or indirectly. The authors should provide a rationale of using whole-body KO, rather than skin-specific KO mice for their study.

3. The authors analyzed Corin function in the sweat duct. But the Corin cascade is highly expressed in the secretory portion. The study will be more complete if the authors analyze the biological meaning of Corin expression in the sweat secretory portion.

4. ENaC was reported to be upregulated in Corin KO kidneys, which was responsible for sodium retention. However, ENaC expression level in Corin KO sweat glands was comparable with WT controls, as shown in Fig 3A and 3B in this manuscript. The authors should quantify the ENaC expression by WB etc to show that ENaC is indeed regulated by Corin in sweat glands.

5. To understand the possible effect of circulating Corin on sweat glands, the authors analyzed heart-specific Corin KO mice. However, Corin is also highly expressed in the kidneys and other tissues. The authors should measure circulating Corin in hcKO mice by ELISA etc to avoid the possibility of "contamination" from non-heart tissues.

6. Figs 2-4: It will be more convincing if the authors show sweat volume in addition to sweat area.

7. Spelling: Page 8, the second line from the bottom, "in" should be changed to "is". Page 5 of the supplemental information, "aimiloride" should be changed to "amiloride". Page 10 of the supplemental information, "injection" should be changed to "injected".

---

## [Decision Letter · Decision Letter 2]

18 Dec 2020

Dear Dr Wu,

Thank you for submitting your revised Short Reports entitled "The protease corin regulates electrolyte homeostasis in eccrine sweat glands" for publication in PLOS Biology. I have now obtained advice from one of the original reviewers and have discussed these comments with the Academic Editor. 

Based on the review, we will probably accept this manuscript for publication, assuming that you will modify the manuscript to address the data and other policy-related requests noted at the end of this email.

We expect to receive your revised manuscript within two weeks.

-  a cover letter that should detail your responses to any editorial requests.

*Published Peer Review History*

*Early Version*

Sincerely,

Ines

--

Ines Alvarez-Garcia, PhD,

Senior Editor,

PLOS Biology

ETHICS STATEMENT:

-- Please include the numbers of the animal care and human tissue license.

-- Please confirm that all clinical investigation have been conducted according to the principles expressed in the Declaration of Helsinki.

Fig. 2B, D, G-J; Fig. 3C-F, I-L; Fig. 4A-D; Fig. S3C, G and Fig. S4A-F 

BLURB

Please also provide a blurb which (if accepted) will be included in our weekly and monthly Electronic Table of Contents, sent out to readers of PLOS Biology, and may be used to promote your article in social media. The blurb should be about 30-40 words long and is subject to editorial changes. It should, without exaggeration, entice people to read your manuscript. It should not be redundant with the title and should not contain acronyms or abbreviations. For examples, view our author guidelines: https://journals.plos.org/plosbiology/s/revising-your-manuscript#loc-blurb

Reviewers' comments

Reviewer #3:

Thank you for addressing all my comments.

---

## [Editor Report · Decision Letter 3]

4 Jan 2021

Dear Dr Wu,

On behalf of my colleagues and the Academic Editor, Cecilia Lo, I am pleased to say that we can in principle offer to publish your Short Report entitled "The protease corin regulates electrolyte homeostasis in eccrine sweat glands" in PLOS Biology, provided you address any remaining formatting and reporting issues.

Before your manuscript can be formally accepted, you will need to complete some formatting- and/or reporting-related requests that will be detailed in an email that will follow this letter. Please note that it usually takes a 2-3 business days for you to receive this email; during this time no action is required by you. Please note that your manuscript will not be formally accepted and scheduled for publication until you have made the required changes.

In the meantime, please log into Editorial Manager at http://www.editorialmanager.com/pbiology/, click the "Update My Information" link at the top of the page, and update your user information to ensure an efficient production process.

PRESS

Thank you again for supporting Open Access publishing. We look forward to publishing your paper in PLOS Biology. 

Sincerely, 

Ines

--

Ines Alvarez-Garcia, PhD 

Senior Editor 

PLOS Biology
